# Changes in Serotonin Modulation of Glutamate Currents in Pyramidal Offspring Cells of Rats Treated With 5-MT during Gestation

**DOI:** 10.3390/brainsci10040221

**Published:** 2020-04-08

**Authors:** Gustavo Hernández-Carballo, Evelyn A. Ruíz-Luna, Gustavo López-López, Elias Manjarrez, Jorge Flores-Hernández

**Affiliations:** 1Instituto de Fisiología Benemérita Universidad Autónoma de Puebla, Puebla C.P.72570, Mexico; l.gustavoh.carballo@gmail.com (G.H.-C.); evy_0927@hotmail.com (E.A.R.-L.); eliasmanjarrez@gmail.com (E.M.); 2Facultad de Ciencias Químicas Benemérita Universidad Autónoma de Puebla, Puebla C.P.72570, Mexico; jose.lopez@correo.buap.mx

**Keywords:** neurotransmitters, development, neurodevelopment, neuromodulation, 5-HT, hyperserotonemia, 5-MT, glutamate, NMDA, ionic currents

## Abstract

Changes in stimuli and feeding in pregnant mothers alter the behavior of offspring. Since behavior is mediated by brain activity, it is expected that postnatal changes occur at the level of currents, receptors or soma and dendrites structure and modulation. In this work, we explore at the mechanism level the effects on Sprague–Dawley rat offspring following the administration of serotonin (5-HT) agonist 5-methoxytryptamine (5-MT). We analyzed whether 5-HT affects the glutamate-activated (I_Glut_) and N-methyl-D-aspartate (NMDA)-activated currents (I_Glut_, I_NMDA_) in dissociated pyramidal neurons from the prefrontal cortex (PFC). For this purpose, we performed voltage-clamp experiments on pyramidal neurons from layers V-VI of the PFC of 40-day-old offspring born from 5-MT-treated mothers at the gestational days (GD) 11 to 21. We found that the pyramidal-neurons from the PFC of offspring of mothers treated with 5-MT exhibit a significant increased reduction in both the I_Glut_ and I_NMDA_ when 5-HT was administered. Our results suggest that the concentration increase of a neuromodulator during the gestation induces changes in its modulatory action over the offspring ionic currents during the adulthood thus contributing to possible psychiatric disorders.

## 1. Introduction

The study of neuromodulators and their influences on voltage-gated and synapse-driven currents are of great interest [1,2,3,4,5,6,7]. In particular, 5-HT catches our attention for its wide distribution on the brain, as well as its role in a large number of behaviors that range from learning and memory to social behavior [8,9,10]. Also, increasing evidence points to alteration of 5-HT signaling as a crucial development component of multiple psychiatric conditions such as schizophrenia, mood disorders, anxiety, and depression [11,12,13,14].

Neuromodulators play an orchestrator role in cellular proliferation, migration, and differentiation during gestational development, giving shape to neuronal communication and circuits [15,16,17,18]. In this context, the study of the 5-HT system is highlighted due to its early development, which is evident from the fifth week of gestation in humans [19] and GD12 in rodents [20,21]. However, the expression of receptors and other serotoninergic structures before the development of its innervation [22] suggests the existence of an exogenous source of 5-HT, which is maternal–placental [23].

Several works have proposed that altering serotonergic homeostasis during offspring’s pre- and post-natal development results in a predisposition to psychiatric disorders [24,25,26,27,28]. Studies where the concentration of 5-HT is alterd in sensitive or critical periods of fetal development show behavioral, cognitive, and emotional abnormalities [29,30], as well as changes in dendritic complexity [31,32] and cellular survival [33] and migration [34,35]. In particular, the artificial increase in the concentration of serotonin using 5-HT transporters (SERT) or monoamine oxidase (MAO) inhibitors, and 5-HT agonists such as 5-MT, induces several brain disorders, for instance, affective-abnormalities, cognitive and fear extinction deficiencies, anxiety-like behaviors [36,37], changes in the expression of 5-HT receptors [38,39,40], abnormalities in the laminar and columnar organization [41,42,43] and the excitability of areas such as the PFC [44].

Particularly, the PFC is densely innervated by axons from the main monoaminergic nuclei, such as the ventral tegmental area and raphe nuclei, which in turn receive projections from the deep layers of the PFC as do multiple cortical and subcortical regions [45]. For this reason, alteration or failure of the processes that regulate the development of proper PFC functioning results in reduced intercommunication within areas [46,47,48], as well as changes in the chemical balance that contributes to multiple cognitive deficits observed in various neuropsychiatric disorders [49].

Through the use of optogenetic inhibition of layer V pyramidal neurons on the valproic acid model [50] or the construction of genetic co-expression networks [51], studies show that neurons in the prefrontal cortex found in layers V–VI are related to the etiology of autism, identifing the fetal stage as the time of greatest sensitivity to the mutations responsible of the disorder.

Based on the foregoing, this work aims to demonstrate the effect of 5-HT on I_NMDA_ and I_Glut_ in dissociated pyramidal neurons of the offspring of rats treated with a serotonergic agonist during pregnancy. We demonstrate that treatment has a controlling influence on the modulatory action of 5-HT on I_NMDA_ and I_Glut_.

We consider that our work contributes to the understanding and interpretation of the effect of maternal stress and inflammation during pregnancy since both increase the placental output of 5-HT to the fetal brain [27,52,53,54,55]. An increasing amount of evidence associates the use of selective 5-HT reuptake inhibitors (SSRIs) during pregnancy with changes in brain morphology and neurodevelopmental disorders [56,57,58]. In the present study, 5-MT, a non-selective serotonin agonist, was applied to pregnant rats in order to examine whether or not such treatment alters the offspring’s serotonergic modulation of I_NMDA_ and I_Glut_.

Our results indicate that the offspring from mothers treated with 5-MT during gestation exhibit a statistically significant increase in the 5-HT inhibitory effect on the I_NMDA_ and I_Glut_ amplitude. Our work is relevant because it shows for the first time that the synapse-driven currents in the brains of offspring, whose mothers were treated with neuromodulators during the gestation, are affected by the same neuromodulators in a different way as the offspring whose mothers were not treated with such neuromodulators. Moreover, our results could lead to the development of new fields in the study of other currents and experimental models.

Furthermore, studies like this are necessary because they offer evidence that alterations in 5-HT levels during gestation lead to severe pathologies in the neural circuits of the offspring. Such is the case of an autism hypothesis [59]. However, works pointing to this hypothesis have only analyzed it at a behavioral level, not at the level of electrophysiological mechanisms.

## 2. Materials and Methods

All protocols and procedures employed in this study were reviewed and approved by the institutional board of bioethics (VIEP/1944/2016) and followed the national (NOM-062-ZOO-1999) and international guidelines of care and use on experimental animals.

### 2.1. Animals

We used pregnant Sprague–Dawley dams with GD0 to GD8 obtained from the vivarium “Claude Bernard” from the Benemérita Universidad Autónoma de Puebla. Dams were housed individually at a temperature/humidity controlled environment in a 12 h light and dark cycle, with ad libitum access to water and food and were assigned to a group using a double-blind design.

For the first part of this study, we used three groups. The first group (5-MT Pre-Post) follows procedures described for Whitaker–Azmitia [59] in order to reproduce the developmental hyperserotonemia (DHS) model of autism. From GD11 to delivery, pregnant dams received a single daily subcutaneous injection of 1 mg/kg 5-MT (non-selective 5-HT agonist) dissolved in the vehicle (0.85% saline with no more than 0.4% dimethylsulfoxide). After birth, pups were treated with 5-MT at the same dose (1 mg/kg) until weaning (postnatal day 24; PND24), thereupon only the male offspring were kept for further electrophysiological analysis in order to reduce the effects of hormonal-cyclic periods in females. The second group (5-MT Gest) follows the same methodology without the administration of pups after delivery. Finally, mothers from the third group (Control) were injected only with the vehicle as well as the pups.

After the first experiment, only the 5-MT Gest and Control groups were maintained. For this part of the study, only the dams were treated in both the 5-MT Gest and the Control groups.

For each electrophysiological protocol, at least three offspring from no less than two different litters were used.

### 2.2. Reagent

Unless otherwise stated, all reagents purchased were from Sigma-Aldrich (St. Louis, MO, USA).

### 2.3. Acute-Dissociation Procedure

PFC neurons from young adults (PND40-PND60) were acutely-dissociated using procedures similar to those described previously [60,61,62]. Dissection of the PFC was limited to V–VI layers, where is the highest density of serotoninergic receptors [63].

Slices were maintained between one and six hours at room temperature (20–22 °C) in Earle’s balanced salt solution (EBSS), buffered with sodium bicarbonate (NaHCO_3_), supplemented with 1 mM pyruvic acid, 0.005 mM glutathione, 0.1 mM NG-nitro-L-arginine and 1 mM kynurenic acid, and bubbled with 95% O_2_/5% CO_2_. The pH was adjusted to 7.4 with sodium hydroxide (NaOH), and osmolarity adjusted to 300 mOsm/L.

After at least one hour of incubation, the PFC slices were prepared for enzymatic treatment. Each slice then was placed in a culture chamber containing 40 mL of Hank’s balanced salt solution (HBSS) mixed with 0.75 mg/mL of papain, buffered with 4-(2-hydroxyethyl)-1-piperazineethanesulfonic acid (HEPES), bubbled with O_2_ and maintained at 35 °C for ten minutes. The solution was supplemented in the same way as the EBSS.

After enzymatic digestion, the tissue was washed with a solution of isethionate and later mechanically separated with various sizes of fire-polished Pasteur pipettes. Suspensions of cells were seeded in 35-mm polystyrene Petri dishes (Corning Inc., Corning, NY, USA) mounted on the recording chamber coupled to an inverted microscope. After ten minutes of incubation, the suspension was washed with a background solution containing 140 mM NaCl, 23 mM glucose, 15 mM HEPES, 2 mM KCl, 2 mM MgCl_2_, 1 mM CaCl_2_ and 1% phenol red, bubbled with O_2_ pH was adjusted to 7.4 with NaOH and osmolarity adjusted to 300 mOsm/L to prepare the tissue for subsequent recording using the voltage-clamp technique.

### 2.4. Whole-Cell Patch-Clamp Technique

Whole-cell voltage-clamp was used to record I_Glut_ /I_NMDA_ in dissociated pyramidal-neurons from the PFC. Recording electrodes were pulled from borosilicate capillary tubes (1B120F-4, WPI, Sarasota, Florida, USA) with a micropipette puller (P-97, Sutter Instruments, CO, USA) and a resistance ranging from 4 to 8 MΩ.

The internal solution consisted of 175 mM N-methyl-D-glutamine (NMDG), 40 mM HEPES, 2 mM MgCl_2_, 10 mM ethylene glycol-bis (β-aminoethyl ether)-N, N, N’, N’- tetra acetic acid (EGTA), 12 mM phosphocreatine, 3 mM Na_2_ATP, 0.35 mM Na_3_GTP and 0.1 mM leupeptin, adjusted to a pH of 7.3 with H_2_SO_4_ / NMDG and 265–270 mOsm/L. The external solution consisted of 127 mM NaCl, 20 mM CsCl, 5 mM BaCl_2_, 2 mM CaCl_2_, 12 mM glucose, and 10 mM HEPES, adjusted to a pH of 7.4 with 300–305 mOsm/L NaOH. Additionally, 10µM glycine was added to the external solution as an NMDA co-agonist for channel activation by glutamate or NMDA [64]. Recordings were obtained with an Axopatch 1-D voltage-clamp amplifier (Molecular Devices, Sunnyvale, CA, USA), controlled with a pCLAMP version 9 (Molecular Devices, Sunnyvale, CA, USA) and a Digidata 1322A digitizer (Molecular Devices, Sunnyvale, CA, USA).

Once the seal was broken, only cells with an input resistance (R_IN_) of less than 25 MΩ were included in the study. The holding potential was set at -80 mV to record I_Glut_ /I_NMDA_. Potassium was blocked by the Cs^+^ and Ba^++^ present in the external solution.

### 2.5. Drug Application

We induced the I_Glut_ and I_NMDA_ currents by applying the compounds using a system of two capillaries placed at 45° to each other and a distance of 300 ± 100 µm from the recorded cell. One capillary contained the external solution with or without the tested neuromodulator (External ± NM), and the other had External ± NM and glutamate or NMDA as required (Figure 1). The solution was changed using solenoid valves (98302-00, Cole-Parmer) controlled by the digital output of the Digidata 1322A system and a control apparatus designed in our laboratory.

During each protocol, we maintained the potential at −80 mV, and we recorded the activity at a sampling frequency of 403.23 Hz (every 2.48 ms). Each sweep lasted six seconds during which the cell was constantly perfused with the external solution with a three-second interruption during which external solution + glutamate (or NMDA) was applied. This condition was maintained until three or four stable current traces were obtained. This same procedure was then repeated by adding the neuromodulator of interest to the external solution. The condition without the neuromodulator was considered as a reference and washing of the effect of the neuromodulators on the current.

We randomly exposed each cell to different neuromodulators or concentrations used in the assigned electrophysiological protocol in order to balance the time-course and cell-viability-induced changes.

### 2.6. Statistical Analysis

We obtainded the neuromodulators effect (5-HT or Mg^++^) as the reduction in the amplitude of the peak or steady-state (SS) of the I_Glut_ or I_NMDA_ according to the control and wash conditions. We calculated the percent reduction as follows:(1)$$\%\;Reduction=\left(\;1-\frac{\delta IL}{\left(\;\left(\delta IR+\delta IW\right)/2\right)}\right)\times 100$$
where δIR is the I_Glut_ or I_NMDA_ density without the neuromodulator (reference condition), δIW is the I_Glut_ or I_NMDA_ density in the wash condition, and δIL is the I_Glut_ or I_NMDA_ density in the presence of the neuromodulator tested. We report values as the mean ± standard error of the mean (SEM). We generated curve fits in Origin 9.1 (Microcal Software Inc., North Hampton, MA, USA) and compared with *F*-test in the form:(2)$$F=\frac{\left(\left(S_{M}-\left(S_{C}+S_{T}\right)\right)/k\right)}{\left(\left(S_{C}+S_{T}\right)/\left(N_{C}+N_{T}-2k\right)\right)},$$
where S_M_ is the residual sum of squares (RSS) for the fit of the combined data of both models, S_C_ is the RSS for the fit of the control group, S_T_ is the RSS for the fit of the tested group, N_C_ and N_T_ is the number of observations for the control and the tested group, respectively, and *k* is the number of parameters of the fit. This statistic test follows the F distribution with *k* and N_1_ + N_2_ − 2*k* degrees of freedom. The statistical comparisons were performed with the help of the extension package *drc* [65] for the statistical environment *R* [66].

We performed the data analysis using a Student’s *t*-test with Welch’s correction or analysis of variance (ANOVA) followed by a multiple comparison test, as appropriate, with the significance set at *p* < 0.05.

## 3. Results

We treated a total of 34 pregnant rats for all the experiments presented in this manuscript, 16 with vehicle only and 18 with 5-MT. From these dams, we used 56 pups to obtain a total of 123 cells: 68 belong to the 16 offspring of the control group, 55 cells from the 14 offspring of mothers treated with 5-MT Gest treatment, and 13 cells belong to 4 with 5-MT Pre-Post treatment. The sample only included recorded neurons with a membrane resistance ≥1 GΩ and an input resistance <25 MΩ. The average of the input resistance for the 123 cells was 17.66 MΩ, with a standard deviation (SD) of 5.42 MΩ.

We determined the cells’ size by measuring the cell capacitance and using a capacitance/area ratio of 1 µF/cm^2^. The average cell capacitance for the control group was 19.46 and 19.03 pF for the 5-MT Gest, with an SD of 4.92 and 5.63 pF, respectively, with no significant difference between groups (t_108.11_ = 0.45, *p* = 0.65). We only compared cells of the 5-MT Pre-Post group (*n* = 13, average = 14.55 pF, SD = 4.18 pF) with those of the control and 5-MT Gest groups used in the same experiment (10 and 12, respectively) because of the small number of cells of this group. No significant difference between the groups was found in this comparison (one way ANOVA F_2,32_ = 1.14, *p* = 0.33).

In each experiment, we evaluated cellular viability with a voltage-clamp ramp (from −100 to +40 mV) lasting 300 ms to generate Na^+^ (I_Na_+ > 1 nA) and Ca^++^ (I_Ca_++ > 100 pA) currents. We excluded any cell with values lower than these from the analysis.

### 3.1. 5-MT Treatment from GD11 to GD21 Increases the 5-HT Effect on I_Glut_

To test the hypothesis that the alteration of 5-HT concentrations of mothers during pregnancy is capable of changing the way 5-HT modulates I_Glut_ of offspring in a life-lasting way, we examined the serotonergic modulation of I_Glut_ in acutely dissociated PFC pyramidal neurons of 40 days old offspring from Sprague–Dawley rats treated with the 5-HT agonist 5-MT from GD11 to GD21 and from GD11 to PND21.

To initiate, we compared a gestational-only treatment (GD11–GD21; “5-MT Gest”; *n* = 10) with the original DHS model (GD11–PND21; “5-MT Pre-Post”; *n* = 13) and a control group (*n* = 12). In particular, we examined whether the 5-MT treatment affects the 5-HT modulation of the I_Glut_ and, whether the gestational treatment alone is capable of producing a statistically significant difference from the control. Moreover, we conducted a one-way between-groups ANOVA to compare the effects of treatments on I_Glut_ (100 µM) in the presence of 5-HT (30 µM). The peak amplitude and SS of I_Glut_ were analyzed (Figure 2), finding significant differences of 5-HT effect on both peak amplitude (F_2,32_ = 16.18; *p* < 0.001) and SS (F_2,32_ = 17.88; *p* < 0.001). Post-hoc comparisons using the Tukey test (*p* < 0.05) indicated that both 5-MT Gest and 5-MT Pre-Post groups are significantly different from the control but no significantly different from each other (Table 1). Altogether, these results suggest that the treatment with 5-MT during an early developmental stage, particularly the period between GD11 and GD21, produces a significant change in the I_Glut_ during the presence of 5-HT.

### 3.2. 5-HT Reduces the NMDA Receptor-Mediated Component of I_Glut_

As GD11–GD21 treatment showed no difference from treatment continued up to PND21, we chose to focus on the former to study how altering the mother’s serotonergic system at a critical period of offspring development can produce lifelong changes in them. Thus, we analyzed the 5-HT (30 µM) effect on both peak amplitude and SS of I_Glut_ (100 µM) for control and 5-MT Gest groups (Control, *n* = 9; 5-MT Gest, *n* = 9). We observed that the peak amplitude shows inhibition of 12.08 ± 1.75% in the control group, and 24.48 ± 2.46% in those offspring from treated mothers, while the SS was reduced 18.83 ± 3.32% and 32.14 ± 4.03%, respectively. The difference between groups was significant in both parameters (Peak, t_14.43_ = 4.108, *p* = 0.001; SS, t_15.44_ = 2.55, *p* = 0.02). During the same protocol, we perfused the cell with Mg^++^ (1 mM), a known NMDA blocker in order to study the remaining glutamate current and the effect of 5-HT on it. In this way Mg^++^ alone reduced the peak amplitude by 94.01 ± 1.09% for the control group and 94.68 ± 0.82% for the treated one, while the SS reduced by 91.31 ± 2.22% and 91.04 ± 0.97% respectively (Figure 3), showing no significant difference between the groups (Peak, t_14.94_ = 0.49, *p* = 0.63; SS, t_10.92_ = −0.11, *p* = 0.911). Similarly, when Mg^++^ was co-applied with 5-HT, neither peak amplitude (Control: 94.35 ± 0.92; 5-MT Gest: 94.74 ± 0.95) nor SS (Control: 92.38 ± 1.87; 5-MT Gest: 91.95 ± 0.82) showed significant differences (Peak: t_15.99_ = 0.29, *p* = 0.78; SS: t_10.92_ = −0.21, *p* = 0.84). It should be noted that no difference was found between the presence and absence of Mg^++^ for the same group in either peak amplitude (Control: t_15.59_ = −0.24, *p* = 0.82; 5-MT Gest: t_15.71_ = −0.04, *p* = 0.97) or SS (Control: t_15.56_ = −0.36, *p* = 0.72; 5-MT Gest: t_15.56_ = −0.72, *p* = 0.48). In summary, the above suggests that treatment with 5-MT causes an increase of up to twice the inhibitory effect of 5-HT over the entire current, mainly affecting the modulation of the NMDA receptor-mediated component of I_Glut_.

To verify that the I_Glut_ component affected by treatment was NMDA receptor-mediated, we compared I_Glut_ and I_NMDA_ in the presence of 5-HT in the same cell. 5-MT treatment caused a significant increase of the 5-HT inhibitory effect on the peak amplitude of both I_Glut_ (t_−3.52_ = 7.64, *p* = 0.008) and I_NMDA_ (t_−3.74_ = 8.15, *p* = 0.005) relative to the control group. Similarly, the serotonergic inhibition of SS was increased on both currents by the treatment (I_Glut_: t_6.43_ = −2.59, *p* = 0.04; I_NMDA_: t_5.09_ = −3.07, *p* = 0.02). When 5-HT (30 µM) was present in the solution, I_Glut_ (100 µM) reduced 20.88 ± 5.01% of the peak amplitude and 29.90 ± 6.04 of the SS in the control group, while in the 5-MT Gest group, the reduction was 42.73 ± 3.65% and 47.87 ± 3.41, respectively. Similarly, I_NMDA_ (100 µM) peak amplitude reduced 22.3 ± 4.43% in the control group and 43.58 ± 3.58% in the treated one, while SS reduced 29.02 ± 6.57 and 50.56 ± 2.43, respectively, in the presence of 5-HT (Figure 4). Additionally, the inhibition caused by 5-HT in I_Glut_ and I_NMDA_ in the same cell exhibits no statistically significant difference between them neither on peak amplitude (Control, t_4_ = 0.52, *p* = 0.63; 5-MT Gest, t_5_ = 0.24, *p* = 0.82) or SS (Control t_4_ = −0.19, *p* = 0.86; 5-MT t_5_ = 0.7153, *p* = 0.51). That is, the treatment affects the 5-HT modulation of both I_Glut_ and I_NMDA_ in the same proportion.

Once we identified that the NMDA receptor-mediated current was the I_Glut_ component mainly affected by the treatment, we wanted to verify that the observed changes were due to alterations in 5-HT modulation of the current and not in the I_NMDA_ itself. To do this, we made an NMDA dose-response curve from 0.3 to 300 µM in both groups. We analyzed the peak amplitude and SS of currents from each concentration (Table 2), then fitted data to a logistic curve (Figure 5) and compared them with an *F*-test. No significant differences were found between groups on peak (F_4,119_ = 1.98, *p* = 0.10) or SS (F_4,119_ = 0.99, *p* = 0.42).

### 3.3. 5-MT Treatment Increases the Modulatory Effect of 5-HT on I_NMDA_

Subsequently, since we did not find differences between groups in I_NMDA_, we made a dose-response curve for the effect of 5-HT (ranged from 1 to 100 µM) in the I_NMDA_ (100 µM). For this curve, we used a total of 54 cells, 33 for the control group, and 21 for the treatment group, each exposed to a maximum of three concentrations of 5-HT administered in random order. The peak amplitude and SS were analyzed (Table 3), itted to a logistic curve with minimum value fixed to zero (Figure 6), and then compared with an F-test. In this way, we found a statistically significant difference between groups in both peak amplitude (F_3,125_ = 10.36, *p* < 0.001) and SS (F_3,125_ = 4.23, *p* = 0.007). Also, the difference between groups for each concentration was measured, finding a significant increase of 5-HT inhibition of I_NMDA_ in treated group related to the control, in 30 µM (t_25.57_ = −4.42, *p* < 0.001) and 100 µM (t_23.01_ = −2.33, *p* = 0.03) for peak amplitude, and 1 µM (t_15.48_ = −2.88, *p* = 0.01), 30 µM (t_26.74_ = −2.07, *p* = 0.048), and 100 µM (t_22.14_ = −2.18, *p* = 0.04) for SS.

## 4. Discussion

This study aims to evaluate the effect of 5-HT on I_NMDA_ and I_Glut_ in dissociated pyramidal neurons of PFC layer V-VI in the offspring of rats treated with a 5-HT agonist during pregnancy. Our data showed that offspring from dams treated with the serotonergic agonist 5-MT (sc. 1 mg/kg/day) from GD11 to G21 presents significant differences on peak amplitude and SS of I_Glut_ in the presence of 5-HT relative to the control group, but did not show differences with treatment continued up to PND21. Observed differences in I_Glut_ in the presence of 5-HT are due to an inhibition of the NMDA receptor-activated component of I_Glut_. Nonetheless, I_NMDA_ itself is not affected except for its serotonergic modulation. In particular, on the 5-HT dose-response curve, an increase in the inhibitory effect of 5-HT on I_NMDA_ was observed, with significant differences of 30 and 100 µM for both peak amplitude and SS.

Our work reaffirms the notion that the effect of 5-HT on PFC pyramidal neurons is mainly inhibitory, and further provides evidence that this inhibitory effect modulates the NMDA-activated component of the I_Glut_. After mimicking a rise in 5-HT concentration with 5-MT, we found an increase of the inhibitory effect of 5-HT, even when the agonist was administered from G11 to birth, or if the treatment was continued until PND21. This increase in the inhibitory effect of 5-HT could lead to a decrease in the excitability of pyramidal neurons in the PFC, similar to what Robello et al. [44] found in the infralimbic cortex, after treatment with a SERT antagonist from PND2 to PND21. Robello et al. [44] found an increase in the excitability of pyramidal neurons from the prelimbic cortex; conversely, we did not find any modulation that could reproduce such a response. The difference in treatment period and layers analyzed could explain the discrepancy between both studies, and it could be due to the different expression and distribution of the 5-HT receptors over time, layers, and subfields of the PFC [24,63].

Animal models using 5-MT to mimic an increase of 5-HT concentration during development have found numerous abnormalities in the serotonergic system [39,67,68], the oxytocinergic cell density [39,69], as well as behavioral [70,71] and morphological changes [43] that have been proposed to be involved in the genesis of multiple neuropsychiatric disorders.

An example of the abnormalities explained above is the autism spectrum, which is the result of an increased 5-HT concentration to which the fetus is exposed during development, a hypothesis described by Whitaker-Azmitia [59] and which provides the basis for the development of her DHS model for autism [59]. However, this hypothesis is based mainly on behavioral evidence [59,70,71]. Therefore, our work examines mechanisms in ionic currents and their modulation, thus supporting this hypothesis. Moreover, our findings shed light on the critical time window and underlying changes involved in this disorder for future study.

Another theory, described by Carlson M.L. [72], postulates that autism could be a hypoglutamatergic disorder; this is based on neuroanatomical and neuroimaging studies indicating alterations in regions rich in glutamatergic neurons, as well as the similarity of symptoms observed in autism and those produced by NMDA antagonists in healthy subjects. In support of this theory, variations of the gene encoding the GluN2B subunit, which does not allow for adequate trafficking and expression to the cell surface, have recently been found in some autistic and schizophrenic patients [73,74]. Concerning this, Yuen et al. [75] demonstrated that activation of the 5-HT_1A_ receptor decreases the traffic and expression of the GluN2B subunit. Therefore, an increase in serotonergic activity, as found in our work, could be contributing through an exacerbation of this mechanism in the development of autism and other disorders.

Altered concentrations of other neurotransmitters, such as dopamine and acetylcholine, have also been associated with the development of disorders such as depression and schizophrenia [76,77,78,79,80]. Therefore, experiments performed with these modulators could provide correlates in the modulation of ionic currents following gestational disturbances, which will be addressed on future work. In this sense, our work is a pioneer in the study of the modulation of ionic currents after variations in the concentration of modulators to which the offspring’s brain is exposed during the gestational period.

Our results are consistent with previous studies showing that prenatal exposure to morphine alters the kinetic properties of I_NMDA_ currents in the hippocampus of the rat offspring [81]. In this context, it is possible that the prenatal exposure to any drug alters the neuromodulator actions on I_NMDA_ and I_Glut_, as well as other ligand-gated channels, in a variety of neurons from the offspring’s brain.

## 5. Conclusions

The gestational treatment with a neuromodulator, 5-MT, induces changes in its modulatory function on glutamate currents during adulthood of the offspring. As this increase in modulators, in our case serotonin, can be produced by strong emotional states in certain periods of the gestation this alteration can lead to changes in the way to process information and can be related with pathological psychiatric disorders.

## Figures and Tables

**Figure 1 brainsci-10-00221-f001:**
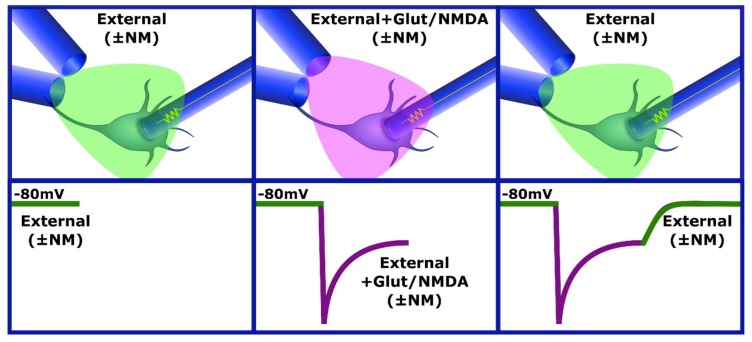
Application of solutions using capillaries. The application of solutions was controlled using solenoid valves; one capillary contained the external solution with or without the tested neuromodulator (External ± NM), and the other had External ± NM and Glutamate or N-methyl–D-aspartate (NMDA) as required. In the bottom panels, there are representative traces of the NMDA-current where the effect of the application of each solution is illustrated.

**Figure 2 brainsci-10-00221-f002:**
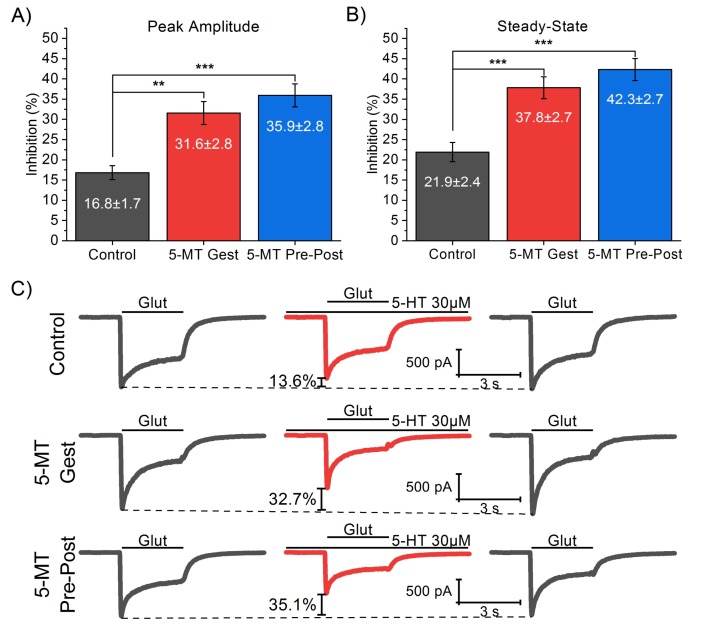
Comparison of the effect of different 5-methoxytryptamine (5-MT) administration times on I_Glut_ in the presence of serotonin (5-HT). We conducted a one-way between-groups ANOVA to compare the effect on I_Glut_ (100 µM) peak amplitude (**A**) and SS (**B**) on the presence of 5-HT (30 µM) in the groups of treatment 5-MT Gest (GD11–GD21; *n* = 10), 5-MT Pre-Post (GD11–PND21; *n* = 13), and control (*n* = 12). Significant differences were found on both current components (Peak amplitude: F_2,32_ = 16.18; *p* < 0.001. SS: F_2,32_ = 17.88; *p* < 0.001) for both treated groups from the control but not between them. (**C**) Representative traces of I_Glut_ in the presence and absence of 5-HT from pyramidal neurons from each group tested. Significant differences among groups are indicated according to the following significance code: *p* < 0.001 (***), *p* < 0.01 (**), *p* < 0.05 (*).

**Figure 3 brainsci-10-00221-f003:**
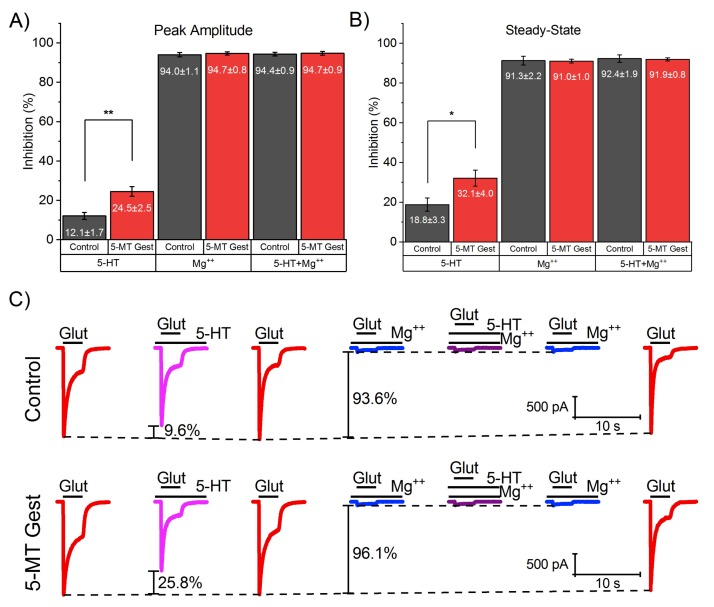
Effect of 5-HT (30 µM) on both peak amplitude and SS of I_Glut_ (100 µM) in the presence and absence of Mg^++^ (1 mM). Peak amplitude (**A**) shows a significant difference (t_14.43_ = 4.108, *p* = 0.001) between inhibition of 12.08 ± 1.75% in the control group (*n* = 9), and 24.48 ± 2.46% in those offspring from treated mothers (*n* = 9) when cells were perfused with 5-HT (t_14.43_ = 4.108, *p* = 0.001), however, no differences were found between the groups when Mg^++^ was applied (t_14.94_ = 0.49, *p* = 0.63) nor when Mg^++^ and 5-HT were co-applied (t_15.99_ = 0.29, *p* = 0.78). Similarly, SS (**B**) shows difference between groups in the presence of 5-HT (Control: 18.83 ± 3.32%; 5-MT Gest: 32.14 ± 4.03%; t_15.44_ = 2.55, *p* = 0.02), but not when it was co-applied with Mg^++^ (t_10.92_ = −0.21, *p* = 0.84) or in the presence of Mg^++^ alone (t_10.92_ = −0.11, *p* = 0.91). Also no difference was found between the presence and absence of Mg^++^ within the same group in either peak amplitude (Control: t_15.59_ = −0.24, *p* = 0.82; 5-MT Gest: t_15.71_ = −0.04, *p* = 0.97) or SS (Control: t_15.56_ = −0.36, *p* = 0.72; 5-MT Gest: t_15.56_ = −0.72, *p* = 0.48). (**C**) Representative traces of I_Glut_ in the presence and absence of 5-HT or Mg^++^ and when co-applied. Significance codes: *p* < 0.001 (***), *p* < 0.01 (**), *p* < 0.05 (*).

**Figure 4 brainsci-10-00221-f004:**
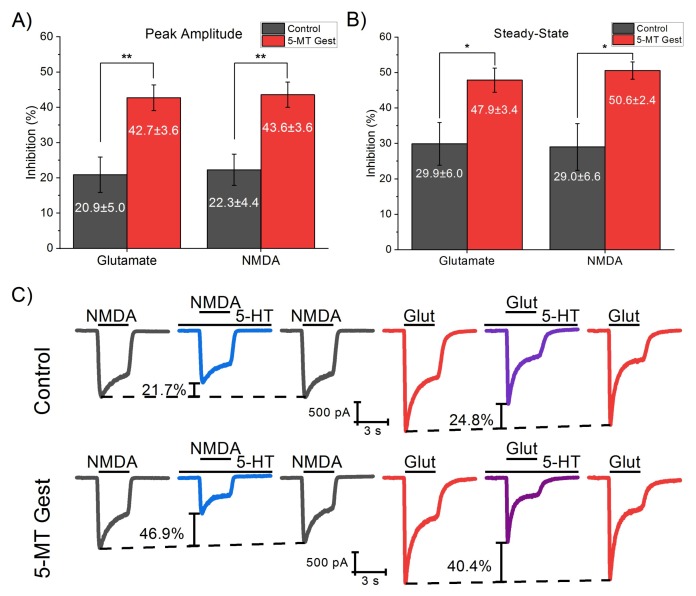
Effect of 5-HT (30µM) on I_Glut_ and I_NMDA_. The 5-MT treatment caused a significant increase of the 5-HT inhibitory effect of the peak amplitude (**A**) of both I_Glut_ (t_−3.52_ = 7.64, *p* = 0.008) and I_NMDA_ (t_−3.74_ = 8.15, *p* = 0.005) relative to the control group. The SS (**B**) presents a similar increase in both currents (I_Glut_: t_6.43_ = −2.59, *p* = 0.04; I_NMDA_: t_5.09_ = −3.07, *p* = 0.02). Conversely, the inhibition caused by 5-HT in I_Glut_ and I_NMDA_ in the same cell presents no statistically significant difference between them neither on peak amplitude (Control, t_4_ = 0.52, *p* = 0.63; 5-MT Gest, t_5_ = 0.24, *p* = 0.82) or SS (Control t_4_ = −0.19, *p* = 0.86; 5-MT t_5_ = 0.7153, *p* = 0.51). (**C**) Representative traces of I_Glut_ and I_NMDA_ both in the presence and absence of 5-HT in the same cell. Significance codes: *p* < 0.001 (***), *p* < 0.01 (**), *p* < 0.05 (*).

**Figure 5 brainsci-10-00221-f005:**
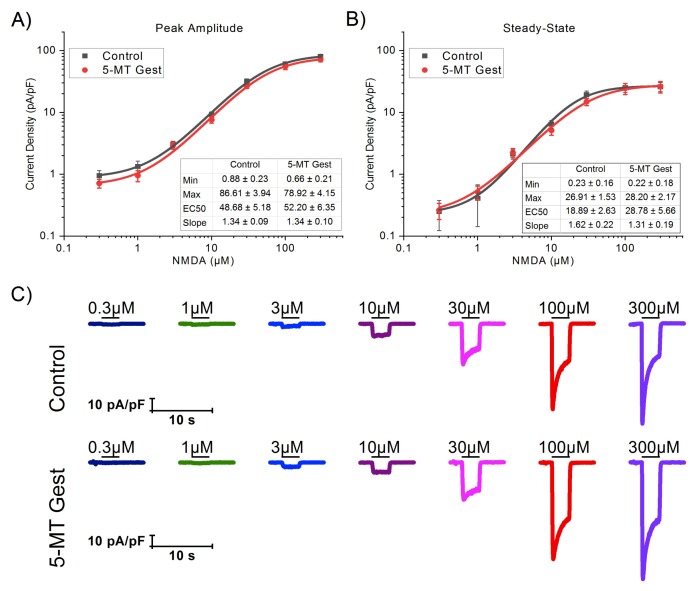
NMDA dose-response curve from 0.3 to 300 µM in both groups. We analyzed the peak amplitude (**A**) and SS (**B**) of currents from each concentration, then fitted the data with a logistic function (solid lines) and compared the fitting curves with an F-test. No significant differences were found between groups (peak amplitude: F_4,119_ = 1.98, *p* = 0.10; SS: F_4,119_ = 0.99, *p* = 0.42). (**C**) Representative traces of I_NMDA_ for each tested concentration on both groups.

**Figure 6 brainsci-10-00221-f006:**
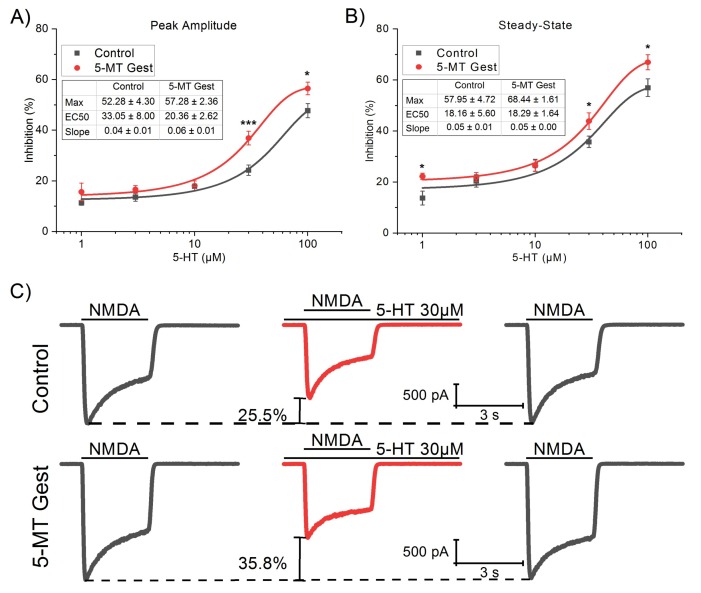
Dose-response curve for the effect of 5-HT from 1 to 100 µM in the I_NMDA_ (100 µM). As with the NMDA dose-response, data were analyzed, fitted (solid lines), and compared with an F-test. In this way, we found significant differences between groups in both the peak amplitude (**A**; F_3,125_ = 10.36, *p* < 0.001) and the steady-state (**B**, F_3,125_ = 4.23, *p* = 0.007). Also, the difference between groups for each concentration was measured, finding significance in 30 µM (t25.57 = −4.42, *p* < 0.001) and 100 µM (t23.01 = −2.33, *p* = 0.03) for peak amplitude, and 1 µM (t15.48 = −2.88, *p* = 0.01), 30 µM (t26.74 = −2.07, *p* = 0.048), and 100 µM (t22.14 = −2.18, *p* = 0.04) for SS. (**C**) Representative traces of I_NMDA_ in the presence and absence of 5-HT 30 µM. Significance codes: *p* < 0.001 (***), *p* < 0.01 (**), *p* < 0.05 (*).

**Table 1 brainsci-10-00221-t001:** Post-hoc comparisons among groups using the Tukey test (*p* < 0.05).

	Peak Amplitude Inhibition (%)				Steady-State Inhibition (%)			
	**MeanDiff**	**SEM**	**q Value**	***p* (<|q|)**	**MeanDiff**	**SEM**	**q Value**	***p* (<|q|)**
**5-MT Gest Vs. Control**	14.73	3.72	5.6	**	15.91	3.79	5.94	***
**5-MT Pre-Post Vs. Control**	19.10	3.48	7.77	***	20.41	3.54	8.15	***
**5-MT Pre-Post Vs. 5-MT Gest**	4.37	3.65	1.69	ns.	4.50	3.72	1.71	ns.

Significance codes: *p* < 0.001 (***), *p* < 0.01 (**), *p* < 0.05 (*), *p* > 0.05 (ns.).

**Table 2 brainsci-10-00221-t002:** Peak amplitude and steady-state of currents from each NMDA concentration.

NMDA(µM)	Peak Amplitude (pA/pF)		Steady-State (pA/pF)	
	**Control**	**5-MT Gest**	**Control**	**5-MT Gest**
**0.3**	−0.95 ± 0.20 (9)	−0.71 ± 0.11 (7)	−0.25 ± 0.13 (9)	−0.26 ± 0.08 (7)
**1**	−1.34 ± 0.30 (9)	−0.95 ± 0.19 (9)	−0.41 ± 0.27 (9)	−0.52 ± 0.13 (9)
**3**	−3.05 ± 0.38 (8)	−2.95 ± 0.40 (9)	−2.13 ± 0.37 (8)	−2.24 ± 0.38 (9)
**10**	−9.26 ± 0.97 (9)	−7.61 ± 0.88 (9)	−6.47 ± 1.00 (9)	−5.12 ± 0.84 (9)
**30**	−31.85 ± 3.08 (9)	−26.74 ± 2.22 (9)	−19.42 ± 2.76 (9)	−14.90 ± 1.95 (9)
**100**	−61.18 ± 4.27 (9)	−55.34 ± 6.05 (9)	−25.25 ± 4.36 (9)	−24.54 ± 5.00 (9)
**300**	−80.80 ± 3.88 (9)	−72.21 ± 6.24 (9)	−26.11 ± 4.35 (9)	−26.20 ± 5.57 (9)

The number of recorded cells for each NMDA concentration and group are displayed between the parentheses.

**Table 3 brainsci-10-00221-t003:** Comparison between groups of the percentage inhibitión of the peak amplitude and steady-state of the I_NMDA_ for each 5-HT concentration.

5-HT (µM)	Peak Amplitude Inhibition (%)			Steady-State Inhibition (%)		
	**Control**	**5-MT Gest**	***p* ( < |t|)**	**Control**	**5-MT Gest**	***p* ( < |t|)**
1	11.33 ± 0.90 (13)	15.68 ± 3.51 (7)	ns.	13.73 ± 2.69 (13)	22.24 ± 1.22 (7)	*
3	13.64 ± 1.65 (9)	16.63 ± 1.57 (13)	ns.	20.44 ± 2.45 (9)	21.97 ± 1.73 (13)	ns.
10	17.88 ± 0.50 (14)	18.05 ± 1.94 (11)	ns.	26.64 ± 2.33 (14)	26.38 ± 2.23 (11)	ns.
30	24.23 ± 2.06 (16)	36.92 ± 2.69 (16)	***	35.77 ± 2.29 (16)	46.94 ± 3.29 (16)	*
100	47.75 ± 2.81 (12)	56.51 ± 2.48 (14)	*	56.99 ± 3.48 (12)	66.98 ± 2.98 (14)	*

The number of recorded cells for each 5-HT concentration and group are displayed between the parentheses. Significance codes: *p* < 0.001 (***), *p* < 0.01 (**), *p* < 0.05 (*), *p* > 0.05 (ns.).

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
