# Peer review of "Changes in Serotonin Modulation of Glutamate Currents in Pyramidal Offspring Cells of Rats Treated With 5-MT during Gestation"

_brainsci, 2020, doi:10.3390/brainsci10040221_

Round 1

Reviewer 1 Report

This study demonstrates that acute administration of 5HT alters Iglut  and INMDA in dissociated PFC neurons of adult male rats and that this alteration is magnified in animals who were exposed to a serotonin agonist (5MT) in utero.

  1. The writing would benefit from overall editing, perhaps with a native English speaker. There are many logical inconsistencies, overstatements and unclear points, along with a few outright typos. For example, the use of phrases like “on the other hand” and “therefore” are not used appropriately.  Also, words like “significative” (used throughout) and ”cero” (line 288) need to be corrected. 

  1. Need more information about the animals - the n provided throughout is the number of cells, but were all the cells from a single animal in each experimental group?Likewise, it is unclear how many offspring per dam were used.  If all the cells within an experimental group came from (even many) offspring of a single dam, then all of the effects reported in the study may actually be due to an abnormality in that one dam.  Finally, if pups were not cross-fostered, then the 5MT injections may have altered maternal behavior, and the effects reported here may be due to serotonin’s effect on maternal behavior rather than a direct effect on the offspring.  These issues should be addressed in a follow-up experiment or at least addressed as major caveats to the current data.

  1. The authors inject pregnant dams, but only use the male offspring with no explanation.Since the serotonin system is known to be sexually dimorphic (indeed, there is at least one study showing a sex difference in 5HT receptors using the DHS model), this decision needs to be explained. 

4.  As the authors note, there have been other studies examining aspects of the developing serotonin system, including of the DHS model (e.g., changes in 5HT receptors).  Therefore, statements that the current work is the only study of mechanism need to be corrected.  Perhaps a further discussion of these other studies could be included. 

Author Response

  1. The writing would benefit from overall editing, perhaps with a native English speaker. There are many logical inconsistencies, overstatements and unclear points, along with a few outright typos. For example, the use of phrases like “on the other hand” and “therefore” are not used appropriately.  Also, words like “significative” (used throughout) and ”cero” (line 288) need to be corrected. 

Response:

We send the manuscript to a native English speaker (Robert Carl Simpson III) correcting the mistakes in English.

  1. Need more information about the animals - the n provided throughout is the number of cells, but were all the cells from a single animal in each experimental group? Likewise, it is unclear how many offspring per dam were used.  If all the cells within an experimental group came from (even many) offspring of a single dam, then all of the effects reported in the study may actually be due to an abnormality in that one dam.  Finally, if pups were not cross-fostered, then the 5MT injections may have altered maternal behavior, and the effects reported here may be due to serotonin’s effect on maternal behavior rather than a direct effect on the offspring.  These issues should be addressed in a follow-up experiment or at least addressed as major caveats to the current data.

Response:

Line 110 For each electrophysiological protocol at least three offsprings from no less than two different litters were used.

Line 197-200 We treated a total of 34 pregnant rats for all the experiments presented in this manuscript, 16 with vehicle only and 18 with 5-MT. From these dams, we used 56 pups to obtain a total of 123 cells: 68 belong to the 16 offspring of the control group, 55 cells from the 14 offspring of mothers treated with 5-MT Gest treatment, and 13 cells belong to 4 with 5-MT Pre-Post treatment.

Although we know that the lack of cross-fostering is a major caveat of our current data, this point was not the objective of our study.

  1. The authors inject pregnant dams, but only use the male offspring with no explanation. Since the serotonin system is known to be sexually dimorphic (indeed, there is at least one study showing a sex difference in 5HT receptors using the DHS model), this decision needs to be explained. 

Response:

Line 102-105 After birth, pups were treated with 5-MT at the same dose (1 mg/kg) until weaning (postnatal day 24; PND24), thereupon only the male offspring were kept for further electrophysiological analysis in order to reduce the effects of hormonal cyclic periods in females.

We restrict this study just to males in order to reduce the effects of hormonal cyclic periods in females. Also, taking account that DHS has being reported as Autism experimental model and the incidence is 4 males - 1 female, we center initially our efforts to the gender with more incidence, males.

  1. As the authors note, there have been other studies examining aspects of the developing serotonin system, including of the DHS model (e.g., changes in 5HT receptors).  Therefore, statements that the current work is the only study of mechanism need to be corrected.  Perhaps a further discussion of these other studies could be included. 

Response:

We agree with reviewer, there are other studies in the developing serotonin systems. We state that the experimental approach centered in electrophysiology of Iglut or INMDA is the first one in the DHS model. We correct.

Line 86 - 87 However, works pointing to this hypothesis have only analyzed it at a behavioral level, not at the level of electrophysiological mechanisms.

Reviewer 2 Report

Minors

1, the title is a little ambiguous. It’s possible for some readers to think that the authors treated the offspring themselves with 5-MT during the gestation. Actually, the authors treated mother animals with 5-MT.

2, line 25, ‘Our results suggests’ should be ‘Our results suggest’

3, line 61, find a better description to replace “has an importance on”

4, line 92 “dissolved on” should be “dissolved in/with”

5, line 144 “respect to” probably refers to “with respect to”

6, in figure 1, it’s better to change “control” in leftmost panel to “vehicle” and change “control” in rightmost panel to “wash-out”. Since the authors also use “control condition” in the statistical analysis function and other figures. There are too many controls. The authors should be better to specify these control conditions. Otherwise, the readers might be confused. It’s difficult for them to distinguish which control is which control. For instance, in the function, the control condition of δIC probably refers to the IGlut /INMDA density without 5-HT. Actually, the vehicle animal group is also control condition.

7, line 201 “on the presence” probably is “in the presence”

8, figure 3, the description of it is too brief and the transition from Glutamate to NMDA in the first paragraph of 3.2 is too hard.

9, line 247 “effect of” to “effect on”

10, figure 4, the authors described that they recorded the IGlut and INMDA in the same cell. The authors should mention whether they took turns to do the recordings to balance the time-course and cell viability induced change. For example, if you have 10 cells, in 5 of them, you should record IGlut first then INMDA. While in the other 5, INMDA were record first and then IGlut.

11, line 270 “To this” to “To do this”

12, in the discussion part, it’s better to deeply discuss the potential mechanism for the phenomena discovered by the authors and propose some reasonable explanations.

Author Response

Minors

1, the title is a little ambiguous. It’s possible for some readers to think that the authors treated the offspring themselves with 5-MT during the gestation. Actually, the authors treated mother animals with 5-MT.

We correct the title

Title Changes in serotonin modulation of glutamate currents in pyramidal offspring cells of rats treated with 5-MT during gestation.

2, line 25, ‘Our results suggests’ should be ‘Our results suggest’

corrected

3, line 61, find a better description to replace “has an importance on”

corrected

Line 70-72 We consider that our work contributes to understand and interpret the effect of maternal stress and inflammation during pregnancy since both increases the placental output of 5-HT to the fetal brain.

4, line 92 “dissolved on” should be “dissolved in/with” corrected

5, line 144 “respect to” probably refers to “with respect to” corrected

6, in figure 1, it’s better to change “control” in leftmost panel to “vehicle” and change “control” in rightmost panel to “wash-out”. Since the authors also use “control condition” in the statistical analysis function and other figures. There are too many controls. The authors should be better to specify these control conditions. Otherwise, the readers might be confused. It’s difficult for them to distinguish which control is which control. For instance, in the function, the control condition of δIC probably refers to the IGlut /INMDA density without 5-HT. Actually, the vehicle animal group is also control condition.

Corrected

Line 156-158 One capillary contained the external solution with or without the tested neuromodulator (External±NM), and the other had External±NM and Glutamate or NMDA as required.

Figure 1 edited

Figure 1: Application of solutions using capillaries. The application of solutions was controlled using solenoid valves; one capillary contained the external solution with or without the tested neuromodulator (External±NM), and the other had External±NM and Glutamate or NMDA as required. In the bottom panels, there are representative traces of the NMDA-current where effect of the application of each solution is illustrated.

Line 167-176 During each protocol, we maintained the potential at -80 mV and we recorded the activity at a sampling frequency of 403.23 Hz (every 2.48ms). Each sweep lasted six seconds during which the cell was constantly perfused with the external solution with a three second interruption during which external solution + glutamate (or NMDA) was applied. This condition was maintained until three or four stable current traces were obtained. This same procedure was then repeated by adding the neuromodulator of interest to the external solution. The condition without the neuromodulator was considered as a reference and washing of the effect of the neuromodulators on the current.

We randomly exposed each cell to different neuromodulators or concentrations used in the assigned electrophysiological protocol in order to balance the time-course and cell viability induced changes.

Line 178-185 We obtained the neuromodulators effect (5-HT or Mg++) as the reduction in the amplitude of the peak or steady-state (SS) of the IGlut or INMDA according to the control and wash conditions. We calculated the percent reduction as follows:

(1)

where δIR is the IGlut or INMDA density without the neuromodulator (reference condition), δIW is the IGlut or INMDA density in the wash condition, and δIL is the IGlut or INMDA density in the presence of the neuromodulator tested. We report values as the mean ± standard error of the mean (SEM). We generated curve fits in Origin 9.1 (Microcal Software Inc., North Hampton, MA) and compared with F-test in the form:

7, line 201 “on the presence” probably is “in the presence” corrected

8, figure 3, the description of it is too brief and the transition from Glutamate to NMDA in the first paragraph of 3.2 is too hard.

We change it

Line 253-265 During the same protocol, we perfused the cell with Mg++ (1mM), a known NMDA blocker in order to study the remaining glutamate current and the effect of 5-HT on it. In this way Mg++ alone reduced the peak amplitude by 94.01±1.09% for the control group and 94.68±0.82% for the treated one, while the SS reduced by 91.31±2.22% and 91.04±0.97% respectively, showing no significant difference between the groups (Peak, t14.94=0.49, p=0.63; SS, t10.92=-0.11, p=0.911). Similarly, when Mg++ was co-applied with 5-HT, neither peak amplitude (Control: 94.35±0.92; 5-MT Gest:94.74±0.95) nor SS (Control: 92.38±1.87; 5-MT Gest:91.95±0.82) showed significant differences (Peak: t15.99=0.29, p=0.78; SS: t10.92=-0.21,p=0.84) . It should be noted that no difference was found between the presence and absence of Mg++ for the same group in either peak amplitude (Control: t15.59=-0.24, p=0.82; 5-MT Gest: t15.71=-0.04, p=0.97) or SS (Control: t15.56=-0.36, p=0.72; 5-MT Gest: t15.56=-0.72, p=0.48). Summaraizing, the above suggest that treatmente with 5-MT causes an increse of up to twice the inhibitory effect of 5-HT over the entire current, mainly affecting the modulation of the NMDA receptor-mediated component of IGlut.

Figure 3: Effect of 5-HT (30µM) on both peak amplitude and SS of IGlut (100 µM) in the presence and absence of Mg++ (1mM). Peak amplitude (A) shows a significant difference (t14.43=4.108, p=0.001) between inhibition of 12.08±1.75% in the control group (n=9), and 24.48±2.46% in those offspring from treated mothers (n=9) when cells were perfused with 5-HT (t14.43=4.108, p=0.001), however, no differences were found between the groups when Mg++ was applied (t14.94=0.49, p=0.63) nor when Mg++ and 5-HT were co-applied (t15.99=0.29, p=0.78). Similarly, SS (B) shows difference between groups in the presence of 5-HT (Control:18.83±3.32%; 5-MT Gest: 32.14±4.03%; t15.44=2.55, p=0.02), but not when it was co-applied with Mg++ (t10.92=-0.21, p=0.84) or in the presence of Mg++ alone (t10.92=-0.11, p=0.91). Also no difference was found between the presence and absence of Mg++ within the same group in either peak amplitude (Control: t15.59=-0.24, p=0.82; 5-MT Gest: t15.71=-0.04, p=0.97) or SS (Control: t15.56=-0.36, p=0.72; 5-MT Gest: t15.56=-0.72, p=0.48). (C) Representative traces of IGlut in the presence and absence of 5-HT or Mg++ and when co-applied. Significance codes: p<0.001 (***), p<0.01 (**), p<0.05 (*).

9, line 247 “effect of” to “effect on” corrected

10, figure 4, the authors described that they recorded the IGlut and INMDA in the same cell. The authors should mention whether they took turns to do the recordings to balance the time-course and cell viability induced change. For example, if you have 10 cells, in 5 of them, you should record IGlut first then INMDA. While in the other 5, INMDA were record first and then IGlut.

Line 174-176 We randomly exposed each cell to different neuromodulators or concentrations used in the assigned electrophysiological protocol in order to balance the time-course and cell viability induced changes.

11, line 270 “To this” to “To do this” corrected

12, in the discussion part, it’s better to deeply discuss the potential mechanism for the phenomena discovered by the authors and propose some reasonable explanations.

Line 399-402. We found that the excess of neuromodulator 5HT, in this case mimicked by 5-MT, generate mother hiperserotoninemia during critical periods of gestation that can induce compensatory changes in the adult life, as the changes in the expression or sensitivity of 5-HT receptors. However, the mechanism by which these compensatory changes can occur remain to be studied.

Reviewer 3 Report

Title:

Changes in serotonin modulation of glutamate currents in pyramidal cells from offspring treated with 5-MT during gestation

Summary:

In this manuscript, the authors utilized a hyperserotonemia rat model developed in previous studies, to show a correlation of 5-MT treatment during gestation and changes of glutamate currents in pyramidal neurons from adult offspring. Specifically, 5-MT treatment during gestation day 11 to 21 was sufficient to lead to increased serotonin-mediated inhibition of glutamate currents, particularly the NMDA receptor-mediated component, in deep layer pyramidal neurons from prefrontal cortex (PFC) of young adult rats. Furthermore, the authors demonstrated that the “increased serotonin-mediated inhibition” phenotype of pyramidal neurons in the treated group was not due to their NMDA response, which was unaffected, but was specifically due to the 5-HT response of these neurons.

Even though the authors provided brief evidence indicating that serotonin signaling is involved in various psychiatric disorders at multiple levels, however, it’s not clear what role(s) the deep layer pyramidal neurons in PFC might particularly play in these disorders. Certainly, there are other brain regions/circuits involved in the etiologies of these disorders. Therefore, in terms of disease modeling, it’s not very convincing to apply the discoveries here to directly interpret any of the psychiatric disorders based on the evidence shown in this manuscript.

Comments:

Major points

As discussed in the “Summary” section, it will be great to see more neuroanatomy/neurobiology experiments to further elucidate the cellular/molecular mechanism(s) underlying the phenotype, or to relate the discoveries to any of the psychiatric disorders mentioned in this manuscript. For instance, 1) the authors didn’t show whether there are differences in 5-HT receptor(s) expression in pyramidal neurons, or changes in cortical structures between control and treated groups, 2) A time-course experiment will help to pinpoint critical stage for phenotype development.

Minor points:

Figure 3 and Figure 4: Why the “inhibition (%)” of peak amplitudes are quite different between Figure 3-> Panel A-> “5-HT” group (12.08±1.75% in the control group and 24.48±2.46% in the treated group) and Figure 4 -> Panel A -> “NMDA” group (22.3±4.43% in the control group and 43.58±3.58% in the treated group). Weren’t cells treated with NMDA + 5-HT the same way?

Line 238-240: Typos in figure legend panel A. “…inhibition of 12.08±1.75% in the control group (n=9), and 24.48±2.46% in those offspring from treated mothers (n=9) during the absence of Mg++, but no significant difference was found between groups when Mg++ was present (t0.29=15.99, p=0.77548).

Line 252-254: Numbers are wrong in the text. “Similarly, INMDA (100μM) peak amplitude reduced 22.3±4.43% in the control group and 43.58±3.58% in the treated one, while SS reduced 29.02±6.57 and 50.56±2.43, respectively, in the presence of 5-HT (Figure 4).”

Author Response

In this manuscript, the authors utilized a hyperserotonemia rat model developed in previous studies, to show a correlation of 5-MT treatment during gestation and changes of glutamate currents in pyramidal neurons from adult offspring. Specifically, 5-MT treatment during gestation day 11 to 21 was sufficient to lead to increased serotonin-mediated inhibition of glutamate currents, particularly the NMDA receptor-mediated component, in deep layer pyramidal neurons from prefrontal cortex (PFC) of young adult rats. Furthermore, the authors demonstrated that the “increased serotonin-mediated inhibition” phenotype of pyramidal neurons in the treated group was not due to their NMDA response, which was unaffected, but was specifically due to the 5-HT response of these neurons.

Even though the authors provided brief evidence indicating that serotonin signaling is involved in various psychiatric disorders at multiple levels, however, it’s not clear what role(s) the deep layer pyramidal neurons in PFC might particularly play in these disorders. Certainly, there are other brain regions/circuits involved in the etiologies of these disorders. Therefore, in terms of disease modeling, it’s not very convincing to apply the discoveries here to directly interpret any of the psychiatric disorders based on the evidence shown in this manuscript.

Line 56-65 Particularly, the PFC is densely innervated by axons from the main monoaminergic nuclei such as the ventral tegmental area and raphe nuclei, which in turn receive projections from the deep layers of the PFC as do multiple cortical and subcortical regions[45]. For this reason, alteration or failure of the processes that regulate the development of proper PFC functioning results in reduced intercommunication within areas [46–48] as well as changes in the chemical balance that contributes to multiple cognitive deficits observed in various neuropsychiatric disorders [49].

Through the use of optogenetic inhibition of layer V pyramidal neurons on valproic acid model [50] or the construction of genetic co-expression networks [51], studies show that neurons in the prefrontal cortex found in layers V-VI are related to the etiology of autism identifing the fetal stage as the time of greatest sensitivity to the mutations responsible of the disorder.

Comments:

Major points

As discussed in the “Summary” section, it will be great to see more neuroanatomy/neurobiology experiments to further elucidate the cellular/molecular mechanism(s) underlying the phenotype, or to relate the discoveries to any of the psychiatric disorders mentioned in this manuscript. For instance, 1) the authors didn’t show whether there are differences in 5-HT receptor(s) expression in pyramidal neurons, or changes in cortical structures between control and treated groups, 2) A time-course experiment will help to pinpoint critical stage for phenotype development.

We agree with reviewer that will be great to see more neuroanatomy/neurobiology experiments to further elucidate the cellular/molecular mechanism(s) underlying the phenotype, or to relate the discoveries to any of the psychiatric disorders mentioned in this manuscript. However, in the manuscript we center our attention in the electrophysiological characterization on glutamate activated currents of the mother hiperserotoninemia, mimicked by 5-MT in a critical period of gestation. We will study some other points of view in future projects.

Minor points:

Figure 3 and Figure 4: Why the “inhibition (%)” of peak amplitudes are quite different between Figure 3-> Panel A-> “5-HT” group (12.08±1.75% in the control group and 24.48±2.46% in the treated group) and Figure 4 -> Panel A -> “NMDA” group (22.3±4.43% in the control group and 43.58±3.58% in the treated group). Weren’t cells treated with NMDA + 5-HT the same way?

We notice this change in 5-HT modulation on cells, they were treated in the same way, we did not find any difference in the treatment or procedure between the first set of data and the posterior. However, the changes in the modulatory action of 5-HT on glutamate or NMDA currents are similar.

In the panel C of figure 3 we correct the label NMDA by Glut.

Line 238-240: Typos in figure legend panel A. “…inhibition of 12.08±1.75% in the control group (n=9), and 24.48±2.46% in those offspring from treated mothers (n=9) during the absence of Mg++, but no significant difference was found between groups when Mg++ was present (t0.29=15.99, p=0.77548). oc

Line 252-254: Numbers are wrong in the text. “Similarly, INMDA (100μM) peak amplitude reduced 22.3±4.43% in the control group and 43.58±3.58% in the treated one, while SS educed 29.02±6.57 and 50.56±2.43, respectively, in the presence of 5-HT (Figure 4).” Corrected
